# Isorhamnetin Suppresses Human Gastric Cancer Cell Proliferation through Mitochondria-Dependent Apoptosis

**DOI:** 10.3390/molecules27165191

**Published:** 2022-08-15

**Authors:** Yehua Li, Baoqiang Fan, Ning Pu, Xue Ran, Tiancheng Lian, Yifan Cai, Wei Xing, Kun Sun

**Affiliations:** 1College of Life Science, Northwest Normal University, Lanzhou 730070, China; 2Cuiying Biomedical Research Center, Lanzhou University Second Hospital, Lanzhou 730030, China

**Keywords:** isorhamnetin, mitochondria, apoptosis, gastric cancer, transcriptome

## Abstract

Derivates of natural products have been wildly utilized in the treatment of malignant tumors. Isorhamnetin (ISO), a most important active ingredient derived from flavonoids, shows great potential in tumor therapy. However, the therapeutic effects of ISO on gastric cancer (GC) remain unclear. Here, we demonstrate that ISO treatment dramatically inhibited the proliferation of two types of GC cells (AGS-1 and HGC-27) both in vitro and in vivo in time- and dose-dependent manners. These results are consistent with the transcriptomic analysis of ISO-treated GC cells, which yielded hundreds of differentially expressed genes that were enriched with cell growth and apoptosis. Mechanically, ISO treatment initiated the activation of caspase-3 cascade and elevated the expression of mitochondria-associated Bax/Bcl-2, cytosolic cytochrome c, followed by the activation of the cleavage of caspase-3 as well as poly ADP-ribose polymerase (PARP), resulting in the severe reduction of the mitochondrial potential and the accumulation of reactive oxygen species (ROS), while pre-treatment of the caspase-3 inhibitor could block the anti-tumor effect. Therefore, these results indicate that ISO treatment induces the apoptosis of GC cells through the mitochondria-dependent apoptotic pathway, providing a potential strategy for clinical GC therapy.

## 1. Introduction

Cancer is a primary cause of death worldwide. In 2020, statistics showed that there were about 19 million new cases and about 10 million deaths caused by cancer worldwide. Among different cancer types, gastric cancer (GC) is a fatal malignancy that causes over 1,000,000 new cases and about 770,000 deaths globally in 2020, exhibiting a high incidence (fifth) and mortality rate (fourth) [1]. Factors such as *Helicobacter pylori* infection, genetic alteration, diet, obesity, and tobacco use are the main causes of GC [2]. Although the mortality of GC has been reduced by treatment options such as surgery, chemotherapy, radiotherapy, targeted therapy, and immunotherapy, the treatment of GC still has serious problems, including drug resistance, poor prognosis, and a high recurrence rate [3]. Thus, novel strategies are needed to alleviate the progression and mortality of GC. Due to the advantages of multi-target, multi-link, and small side effects, natural medicine has recently become an emerging option for clinical anticancer drug investigation [4]. For example, natural product-derived doxorubicin and camptothecin have been widely used as an anticancer drug [5]. Nevertheless, GC is not sensitive to these compounds. Hence, new effective derivates from natural products still need to be developed and tested in GC therapy.

Flavonoids are abundant in daily intakes of food, such as fruits, vegetables, tea, and red wine. Recently, a variety of flavonoid agents have been applied to different human disease therapies, such as cardiovascular disease, breast cancer, and ovarian cancers [6,7,8]. Among these agents, isorhamnetin is a natural product isolated from various plants, including *Hippophae rhamnoides* L., *Ginkgo biloba* L., and *Tamarix ramosissima*, and a kind of immediate metabolite of quercetin in mammals [9,10]. Diverse studies have demonstrated that ISO exhibits remarkable effects on immunomodulatory, anti-inflammatory, and cardiovascular as well as cerebrovascular protection [11,12,13]. Recently, ISO has received attention due to its tumor-suppressing role in different human cancers, including colorectal, skin, lung, and breast cancers [14,15,16,17,18]. In these cancers, ISO exhibits comprehensive anti-tumor activities by repressing cell proliferation and migration and activating apoptosis [19,20,21]. Even though the cytostatic and pro-apoptotic effects of ISO have been extensively studied, the potential effects of ISO in GC therapy and the molecular mechanisms by which ISO induces apoptosis are still unknown.

Apoptosis plays critical functions in controlling cell homeostasis and proliferation. Hence, the aberrant activation of apoptosis has been observed in various diseases and contributes to numerous human cancers [22], emphasizing the activation of apoptosis in cancer therapy [23,24]. Ramachandran et al. reported that ISO could modulate PPAR-γ cascade activity and further inhibit proliferation and invasion, while the use of PPAR-γ inhibitor mildly blocked the effects of ISO [25]. In addition, Lee et al. previously uncovered that ISO activated the apoptosis of LLC cells in part through mitochondria-cytochrome C-caspase-9 cascade [26], and Hu et al. demonstrated that ISO has an important role in ROS-mediating CaMKII/Drp1 signaling by regulating mitochondrial fission and apoptosis [12]. These studies have suggested that the anti-tumor function of ISO is mainly based on the apoptosis-dependent pathway. Our group previously found that ISO treatments could enhance the radiosensitivity and apoptosis of A549 cells through the NF-κB and IL-13 signaling pathways [15]. Collectively, ISO has exhibited pro-apoptotic effects in various tumor therapies.

Due to the profound pro-apoptotic effects of ISO, in this study, we further investigated the anti-tumor activities of ISO in GC cells and revealed the antineoplastic properties of ISO both in vitro and in vivo. In addition to the PPAR-γ activation induced by ISO treatment, we found that ISO could induce high cytotoxicity of GC cells and significantly inhibit cell proliferation and migration. We also revealed that the mechanisms underlying this effect were involved in mitochondria-dependent caspase-3 cascade. Overall, these findings implicate the potential of ISO in gastric cancer treatment and further reveal the primary pathways of ISO-induced apoptosis.

## 2. Results

### 2.1. ISO Induces Apoptosis of GC Cells

In order to evaluate the potential effects of ISO on GC cells, we first tracked the morphology of ISO-treated GC cells. Tubulin-based analyses of cell morphology exhibited significant changes in the two types of GC cells (AGS-1 and HGC-27) during ISO treatment; rounded cells with progressive nuclear shrinkage and vesicle formation were observed (Figure 1A). Severe disruption of the cell morphology indicated that the ISO treatment may have induced cell apoptosis. DNA fragmentation is the hallmark of apoptosis [21]; therefore, we conducted a TUNEL assay to evaluate the effects of ISO on this event. The results show that the ISO treatment increased the proportion of TUNEL-positive cells in a time-dependent manner (Figure 1B,C). The TUNEL fluorescence intensity of AGS-1 cells (Figure 1D) and HGC-27 cells (Figure 1E) had increases of 300–500-fold and 300–800-fold, respectively.

Furthermore, we performed annexin V/PI staining and flow cytometric analysis to detect the apoptosis levels of ISO-treated cells. The staining results show that the ISO treatment drastically increased the proportion of cells in the early apoptotic states at 24 h, while prolonged treatment (48 h) further increased the proportion of cells in the late apoptotic stage (Figure 1F,G). The analysis of the apoptosis rate (assessed by the proportion of all apoptotic cells) further revealed that the ISO treatment induced apoptosis in a time-dependent manner (Figure 1H,I). Based on these results, we tested the anti-tumor effects of ISO in vivo. The results show that the ISO treatment drastically suppressed tumor growth in mice with AGS-1 tumor xenografts (Figure 1J,K).

### 2.2. ISO Induces the Caspase-3 Cascade in GC Cells

It is well established that the cleavage of caspase-3 is the key regulatory event in the activation of apoptotic signaling [27]. Thus, we performed co-staining of caspase-3 and annexin V to detect caspase-3 activity and apoptosis levels. Consistently, caspase-3 and annexin V were both strongly activated after ISO treatment in AGS-1 and HGC-27 cells (Figure 2A,B). Cleaved caspase-3 functions as an executioner caspase and cleaves most of its substrates to induce apoptosis [28,29]. To further confirm the ISO-induced activation of caspase-3 in GC cells, we examined the cleavage level of caspase-3 and PARP. After ISO treatment at 20 μM for 6 h, higher levels of cleaved caspase-3 and cleaved PARP were observed at 48 h (Figure 2C–H).

We then questioned whether the ISO-induced apoptosis was caspase-3 activation-dependent. Thus, we pre-incubated AGS-1 and HGC-27 cells with the caspase-3 inhibitor (Ac-DEVD-CHO) for 30 min, and then treated these cells with ISO for 72 h. The viability of AGS-1 and HGC-27 cells, as measured by the CCK-8 assay, decreased after treatment with ISO (Figure 2I,J), while treatment with the 50 μM Ac-DEVD-CHO alone had little effect on the viability of the GC cells. However, pre-treatment with Ac-DEVD-CHO drastically abolished the ISO-induced suppression in cell viability in an ISO dose-dependent manner; further, 50 μM Ac-DEVD-CHO completely blocked ISO-induced cell apoptosis (Figure 2I,J). Similarly, the anti-tumor effects of ISO were also blocked in vivo by Ac-DEVD-CHO (Figure 1J,K), indicating that the ISO-induced apoptosis activation was caspase-3 dependent.

A previous study demonstrated that ISO modulated PPAR-γ cascade activity and that PPAR-γ inhibitor (GW9662) could partially reverse ISO-induced apoptosis in 12 h [25]. Here, we also involved the treatment of GW9662 in both AGS-1 and HGC-27 cells and compared the block effect with Ac-DEVD-CHO. After 72 h of ISO treatment, we found that the pre-treatment of GW9662 rescued 20% of ISO-induced suppression, which was much weaker than that of the Ac-DEVD-CHO (~70%) (Figure 2I,J). This result shows that the ISO-induced apoptosis activation primarily depended on caspase-3 activation.

### 2.3. ISO Induces the Mitochondrion-Dependent Apoptosis of GC Cells

The mitochondria membrane potential (MMP) has been identified as an important indicator of mitochondrial homeostasis, and mitochondrial depolarization is generally associated with apoptosis [30,31]. To determine whether ISO treatment induced mitochondrion-dependent apoptosis, we first detected the MMP of ISO-treated cells using MitoTracker Red CMXRos. The results show that the MMP gradually decreased with the extension of the ISO treatment duration (Figure 3A,B). The fluorescence intensity of MitoTracker Red CMXRos in AGS-1 cells (Figure 3C) and HGC-27 cells (Figure 3D) was also examined. Similarly, we performed JC-1 staining to detect MMP loss in GC cells under ISO treatment. The mitochondrial depolarization analysis showed that ISO treatment markedly enhanced the mitochondrial depolarization in both AGS-1 (Figure 3E) and HGC-27 cells (Figure 3F). Moreover, we found that ISO treatment upregulated reactive oxygen species (ROS) production in AGS-1 (Figure 3G) and HGC-27 (Figure 3H) cells; this indicates that the ISO treatment impaired mitochondrial homeostasis.

Bcl-2 family proteins, including anti-apoptotic protein Bcl-2 and the pro-apoptotic protein Bax, are major regulators of mitochondrial integrity and apoptosis [32]. Thus, we investigated if these proteins participated in mitochondrial permeability changes. The ISO treatment did not affect Bax or Bcl-2 expression (data not shown). Next, we isolated the mitochondria and examined the expression levels of these proteins. As shown in Figure 3I,J, Bax expression increased markedly during the ISO treatment, which concomitantly repressed the expression of mitochondrion-associated Bcl-2. Consequently, the ratio of Bax/Bcl-2 gradually increased with the ISO treatment in both the AGS-1 and HGC-27 cells (Figure 3K). In drug-induced intrinsic apoptosis signaling, Bax activation contributes to cytochrome c release from the mitochondria and subsequent caspase activation [33]. As expected, with the increase in the duration of ISO administration, we also detected the gradual upregulation of cytochrome c in the total cell lysates (Figure 3I,J,L), which was a likely consequence of the loss of MMP observed in Figure 3A–F. To further clarify that ISO-induced apoptosis relies on Bax activation, we used short hairpin RNA (shRNA) to knock down Bax in both AGS-1 and HGC-27 cells (Figure 3M), and found that Bax knockdown could also block cell ISO-induced apoptosis (Figure 3N).

Collectively, these results demonstrate that ISO-induced apoptosis in GC cells is mitochondrion-dependent and mediated by caspase activation.

### 2.4. Transcriptome Analysis of ISO-Treated GC Cells

To determine the regulatory mechanism underlying the ISO-induced anti-tumor activity in GC cells, we performed RNA-seq of the total RNA from AGS-1 and HGC-27 cells that were treated with 20 μM ISO for 24 h or not. Correlation analysis of RNA-seq revealed that the duplicates were highly correlated (Appendix A), while PCA analysis showed that the ISO treatment resulted in differences in the transcriptomes (Appendix A). In the case of the consistent transcriptome levels (Figure 4A), we identified 465 and 339 differentially expressed genes in the AGS-1 cells (ISO vs. Control) (Figure 4B) and the HGC-27 (ISO vs. Control) cells (Figure 4C), respectively. Heat map analysis was used to depict these differentially expressed genes (Figure 4D and Appendix A). Using gene ontology (GO) analysis, we further found that these differentially expressed genes were enriched in cell growth, apoptosis, migration, and cytoskeleton pathways (Figure 4E,F). Meanwhile, these differentially expressed genes could be classified into three major categories: biological processes, cellular components, and molecular functions (Appendix A).

Taken together, these results indicate that ISO treatment regulates various processes associated with cell homeostasis, including cell growth, apoptosis, and other related biological processes.

### 2.5. ISO Inhibits GC Cell Proliferation

To further assess the ISO-mediated inhibition of human GC cell proliferation, we conducted CCK-8 and colony formation assays using the two types of GC cells. Upon treatment with increasing doses of ISO (0, 10, 20, 50, and 100 μM) and an increasing time course (24, 48, and 72 h), AGS-1 and HGC-27 cells exhibited enhanced growth inhibition (Figure 5A–D). Accordingly, pre-treatment with ISO repressed the colony-formation ability of AGS-1 and HGC-27 cells in a dose-dependent manner (Figure 5E). The colony-formation ability of both AGS-1 and HGC-27 cells pre-treated with 20 μM ISO was inhibited by >50% after 14 days of culture (Figure 5F).

Collectively, these results demonstrate that the ISO treatment suppressed cell proliferation via apoptosis activation.

### 2.6. ISO Suppresses GC Cell Migration

To further evaluate the effect of ISO on the cell migration of GC cells, we performed Transwell and wound-healing analyses of these cells after ISO treatment (20 μM, for 24 h or 48 h). As shown in Figure 6A,B, the ISO treatment distinctly suppressed the migration ability of AGS-1 and HGC-27 cells. The efficiency of this suppression was further calculated by assessing the OD_570_, which indicated a suppression efficiency of >50%. Next, we performed a wound-healing assay via 24-h continuous real-time imaging and analyzed the relative wound gap area at 6, 12, and 24 h. Compared to the case at 0 h, the low-dosage (10 μM) ISO treatment showed a negligible effect on cell migration, whereas a higher dosage (20–100 μM) of ISO strongly inhibited cell migration (Figure 6E–H).

Thus, our results show that ISO treatment notably suppressed the migration of AGS-1 and HGC-27 cells in a dose- and time-dependent manner.

## 3. Discussion

Gastric cancer (GC) is a fatal malignant tumor worldwide. In spite of the advancement in the early detection and chemotherapy of GC, there has been little effect on increasing survival among GC patients. In recent decades, due to the promising efficacy and fewer side effects, derivatives of medicinal natural products exhibit great potential and have attracted extensive attention in cancer therapy. ISO (a kind of immediate 3’-O-methylated metabolite of quercetin) has recently been reported to possess remarkable anti-tumor activity against several human cancer types [19,34,35]. Specifically, ISO repressed TNF-α-induced inflammation and cell proliferation in BEAS-2B cells (human bronchial epithelial cell lines), via the MAPK and NF-κB pathways [36]. Further, our group previously discovered that ISO treatment could enhance the radiosensitivity of lung cancer cells through the IL-13 and NF-κB signaling pathways [14]. However, the anti-tumor effects of ISO on GC cells and how ISO suppresses cell proliferation remain poorly understood.

Cancer cells exhibit common characteristics of infinite proliferation. To this end, we conducted CCK-8 assays to explore the anti-cancer activity of ISO in the two types of GC cells. Interestingly, the ISO was selectively cytotoxic to both types of GC cells in a time- and concentration-dependent manner but exerted no toxic effects on normal human epithelial GES-1 cells under the same conditions. A decrease in the colony formation of GC cells (under ISO treatment) and obvious morphological changes in these cells (condensed nuclear chromatin, vesicle formation, and increased numbers of apoptotic bodies) further confirm the potential tumor-suppressor ability of ISO against GC, which is consistent with the effects of ISO observed in vivo. In cancer therapy, therapeutic efficacy and prognostic survival are mainly relied on in tumor cell movement. In advancing the impact of ISO on cell proliferation inhibition, we also observed that the ISO treatment dramatically mediated the suppression of GC cells’ migration ability in this study, which occurred in an ISO dose-dependent manner.

The RNA-seq data from ISO-treated GC cells yielded genes with differential expressions that were mainly related to cell growth, migration, cytoskeleton, and the mitochondrial membrane, whereas the GO and KEGG analyses of these genes revealed that they were enriched in signaling pathways associated with apoptosis. In addition to apoptosis, the GO analysis also showed that ISO treatment plays an important role in several signaling cascades, including Hippo signaling, tight junctions, MAPK signaling, and ion hemostasis. Nevertheless, the regulatory network for ISO appears to be complicated, and multiple surveillance mechanisms are required to clarify the regulatory effects of ISO on various signaling pathways.

Apoptosis is extremely important in the progression of benign tumors into malignant neoplasms; thus, it is considered a potential target for the treatment of different cancers [37,38,39]. Nucleosome-sized fragments, membrane blebs, and condensed cellular compartments are involved in this complex, sequential process. As expected, ISO-treated GC cells revealed many TUNEL-positive cells, annexin V–FITC, and PI co-staining, further substantiating that the cytotoxicity of the ISO treatment was mediated by the induction of apoptosis. Meanwhile, we observed decreased MMP and JC-1 staining intensities in ISO-treated GC cells; this indicates that the ISO treatment leads to severe mitochondrial dysfunction and suggests that ISO facilitates the apoptosis destiny of GC cells.

The imbalance between the antiapoptotic and proapoptotic Bcl-2 family members facilitates the cytochrome c release from mitochondria and subsequent activation of caspase-9 and -3 and further apoptosis. To further elucidate the role of ISO in inducing apoptosis, the levels of apoptotic signaling proteins (e.g., Bax, Bcl-2, caspase-3) were measured under ISO treatment in AGS-1 and HGC-27 cells. The immunoblotting demonstrated that ISO increased the expression of the mitochondrion-associated Bax, cytoplasmic cytochrome c, cleaved caspase-3, and cleaved PARP proteins, while the expression of mitochondria-associated Bcl-2 protein decreased. These striking changes in the expression levels of these proteins indicate that ISO might contribute to apoptosis activation, thereby giving rise to the anti-tumor effects. A previous study revealed that ISO modulated PPAR-γ cascade activity and induced apoptosis [25]. Here, we compared the effects of PPAR-γ inhibitors and caspase-3 inhibitors in ISO-induced GC apoptosis; the results show that caspase-3 inhibitor pre-treatment completely blocked long-term ISO treatment-induced apoptosis, indicating that ISO treatment initiates a mitochondrion-dependent intrinsic apoptotic pathway. However, apoptosis activation is regulated via several important pathways and mitochondrion-dependent apoptosis is composed of various steps. The precise role of ISO in mitochondrion-related apoptotic activation remains to be determined. In this regard, which specific signaling pathway in ISO-mediated apoptosis activation is necessary should be studied in the future.

In conclusion, this study shows that ISO treatment in GC cells initiated the activation of caspase-3 cascade, the upregulation of cytochrome c, Bax/Bcl-2, and cytosolic cytochrome c, and the cleavage of caspase-3 as well as PARP, resulting in mitochondrial homeostasis imbalance and apoptosis. This effect of ISO is primarily mitochondrial-dependent and contributes to the selective anti-tumor activity in the suppression of the migration and invasion of human GC cells. Collectively, ISO targeting mitochondrion-dependent apoptosis indicates the potential of using ISO as a therapeutic agent for gastric cancer treatment.

## 4. Materials and Methods

### 4.1. Subcutaneous Xenograft Tumor Models

Male nude recipient mice, 4–6-weeks old, were purchased from the SLAC Laboratory Animal Company and raised under pathogen-free conditions. Cells for xenograft (5 × 10^5^) were implanted subcutaneously within a Matrigel matrix (total 0.1 mL) into the mice’s right flanks. At 5 days after the xenograft, ISO (5 mg/kg) treatment was conducted through intraperitoneal injection every 3 days. For apoptosis inhibition, caspase-3 inhibitor Ac-DEVD-CHO (3 mg/kg) was administered every day through intraperitoneal injection after ISO treatment until sacrifice. A vernier caliper was used to measure the tumor volume at the indicated time and the tumor was allowed to grow to reach an average size of 150 mm^3^.

### 4.2. Cell Culture

Human gastric carcinoma AGS-1, HGC-27, and HEK293FT cells were purchased from the Cell Bank of the Chinese Academy of Science. The AGS-1, HGC-27, and HEK293FT cells were cultured with RPMI-1640 (Gibco, Waltham, MA, USA, 11875101) or DMEM (Gibco, 11965092) supplemented with 10% fetal bovine serum (Gibco, 10099) and 1% penicillin–streptomycin (100×) (Yeasen, Shanghai, China, 60162ES76).

### 4.3. Reagents

Isorhamnetin (CAS Number: 480-19-3) was purchased from Baoji Herbest Bio-Tech (Baoji, China). GW9662 (HY-16578) was purchased from MedChemExpress. Caspase-3 inhibitor Ac-DEVD-CHO (C1206), Hoechst 33258 (C1017), and Mito-Tracker Red CMXRos (C1049B) were purchased from Beyotime (Nantong, China). DAPI (D1306) was purchased from Invitrogen. Anti-β-actin (A2228) was from Sigma (St. Louis, MO, USA). Antibodies to Bax (2774), Bcl-2 (D55G8) (Human Specific) (4223) and cytochrome c (4272), cleaved caspase-3 (Asp175) (5A1E) (9664), caspase-3 (9662), PARP (9542), cleaved PARP (Asp214) (D64E10) XP^®^ (5625), PARP (9452), COX IV (4844), and anti-tubulin (2144) were obtained from Cell Signaling Technology.

### 4.4. shRNA Plasmid Construction, Lentivirus Production, and Cell Infection

For shRNA constructs, Bax target sequences and a scrambled sequence were individually cloned into a pLKO.1-TRC vector (AgeI and EcoRI sites).

For lentiviral particle production, 5 μg of shRNA construct, 3.75 μg of psPAX2, and 1.5 μg of pMD2.G were co-transfected into HEK293FT cells (70–80% confluence) in a 6 cm dish. The supernatant containing lentivirus was harvested twice at 48 and 72 h after transfection and further filtered through a 0.45 μm filter before use. To infect the GC cells with lentivirus, they were cultured in a medium containing a lentivirus supplement with 1 mg/mL polybrene (Beyotime, C0351 Sigma).

### 4.5. Cell Morphology Analysis

AGS-1 and HGC-27 cells were seeded on 35-mm glass bottom dishes and incubated overnight. After ISO treatment, the cells were fixed with 4% formaldehyde (15 min) and permeabilized with 0.05% Triton X-100 (5 min). The cells were then blocked with 5% BSA (30 min) and incubated with anti-tubulin (1:2000) for 1 h at room temperature. After washing 3 times with PBS, the cells were incubated with fluorescent secondary antibody (1:1000) for another hour. The nuclei were stained with DAPI (5 min). Fluorescent images were collected using an Olympus FV3000 microscope and processed by ImageJ.

### 4.6. Cell Proliferation Assay

AGS-1 and HGC-27 cells (5 × 10^3^) were seeded in 96-well plates, after treatment with ISO (0, 10, 20, 50, and 100 μM) and Ac-DEVD-CHO (10, 20, and 50 μM) or neither at the indicated time. The DMSO treatment group was set as the control. The cell number was measured by the Cell Counting Kit-8 (Beyotime, C0038) and a microplate reader (Synergy 2 Multi-Mode Microplate Reader, BioTek, Winooski, VT, USA).

### 4.7. Colony Formation Assay

A colony formation assay was performed as described with a few modifications [40]. Cells were seeded on a 60-mm dish (1000 cells per group). After 24 h of incubation, ISO (0, 20, 50 μM) was used to treat the cells for another 24 h, then they were rinsed with PBS, and replaced with fresh medium. After 14 days of incubation, the colonies were fixed in 4% formaldehyde (15 min) and stained with 0.5% crystal violet (30 min), followed by manually counting.

### 4.8. TdT-Mediated dUTP Nick-End Labeling (TUNEL)

AGS-1 and HGC-27 cells were seeded on 35-mm glass bottom dishes and incubated for 24 h. The cells were then fixed in 4% formaldehyde (30 min) and permeabilization with 0.05% Triton X-100 (5 min). After rinsing, the cells were incubated with a One-Step TUNEL Apoptosis Assay Kit (Beyotime, C1086). Images were captured by an Olympus FV3000 microscope and analyzed using Fiji ImageJ.

### 4.9. Apoptosis Assays

After treatment with 20 μM ISO for 24 or 48 h, the cells were digested with trypsin. The cells were then washed with PBS and detected with an Annexin V-FITC Apoptosis Detection Kit (Beyotime, C1062L), followed by measurement with a BD FACS Celesta^TM^ (BD Biosciences, Franklin Lakes, NJ, USA). Caspase-3 and annexin V staining was conducted with (Beyotime, C1077M) according to the protocol. Images were captured by an Olympus FV3000 microscope and further analyzed by Fiji ImageJ.

### 4.10. Mitochondrial Staining of Cells

After treatment with 20 μM ISO, the cells were stained with Mito-Tracker Red CMXRos working fluid (37 °C, 30 min), followed by staining with Hoechst for 10 min. After staining, the cells were washed with PBS and pre-warmed. FluoroBrite DMEM (GIBCO) supplemented with 10% FBS was used for the lice-cell imaging. Images were captured by an Olympus FV3000 confocal microscope.

### 4.11. Mitochondrial Membrane Potential Measurement (MMP)

The MMP was detected as described with a few modifications [41]. After treatment with 20 μM ISO for 24 h, the MMP of the cells was detected using a JC-1 detection kit (BD Biosciences, 551302). The cells were incubated with JC-1 working solution (37 °C, 20 min) and further washed with 1× assay buffer twice, followed by flow cytometry analysis. The statistics of the MMP were measured by the ratio of JC-1 monomers to JC-1 aggregates.

### 4.12. Reactive Oxygen Species (ROS) Assay

After treatment with 20 μM ISO for 24 or 48 h, the cells were digested with trypsin. Then, the cells were stained with DCFH-DA (Beyotime, S0033M) at 37 °C for 20 min, followed by washing with PBS 3 times and flow cytometry analysis.

### 4.13. Cell Mitochondrial Isolation

The mitochondria fraction was collected using a Cell Mitochondria Isolation Kit (Beyotime, C3601) with a few modifications. After treatment with ISO, 5 × 10^7^ cells were harvested with trypsin and washed with PBS. The cells were then homogenized, and the lysates were centrifuged at 1000× *g* (5 min) for the removal of unbroken cells. The supernatant was centrifuged at 12,000× *g* (10 min), and the mitochondria fractions were obtained in the pellets. The pellets were further lysed with 1 × sodium dodecyl sulfate (SDS) and analyzed using Western blotting.

### 4.14. Cell Migration Assay

Cell migration assays were performed using Transwell Permeable Supports (Costar Corning, NY, USA) with a pore size of 8.0 um, according to the Boyden chamber method. AGS-1 and HGC-27 cells seeded on the upper chamber supplemented with RPMI-1640 (serum-free) were pre-treated with 20 μM ISO. The lower chamber was supplemented with normal RPMI-1640 medium. After 24 h of incubation, the cells that migrated into the lower chamber were stained with 0.5% crystal violet (30 min) and observed using an Olympus IX73. The cells were further eluted by 50% ethanol and 0.1% acetic acid and detected by BioTek Synergy NEO at 550 nm.

### 4.15. Wound-Healing Assay

A wound-healing assay was performed as described with a few modifications [42]. AGS-1 and HGC-27 cells were seeded in a 96-well plate overnight before ISO treatment. A sterile pipette tip was used to generate the wounded cell line, and the floating cells were removed using a PBS wash. The wounded cells were cultured for up to 24 h and images were captured by EnSight™ Multimode Microplate Reader (PerkinElmer, Waltham, MA, USA).

### 4.16. Preparation of RNA-seq Libraries and Sequencing

RNA samples were extracted using a TRIzol and QIAGEN RNeasy Mini Kit. The cDNA libraries were prepared by the VAHTSTM mRNA-seq V3 Library Prep Kit (Vazyme, Nanjing, China, NR611-01) and subjected to an Illumina NextSeq sequencer.

### 4.17. Western Blotting

After the indicated ISO treatment, AGS-1 and HGC-27 cells were lysed in 1 × SDS. The protein samples were resolved in 10% SDS-PAGE gel and transferred to PVDF membranes. After blocking with 5% non-fat dry milk, each protein on separate membranes was detected by incubation with the indicated primary antibodies at 4 °C overnight, followed by the corresponding HRP-conjugated secondary antibodies, and then detected using an ECL detection kit and Tanon 5200.

### 4.18. Statistical Analysis

The data are presented as the means ± S.D. in this study. Statistical analyses (two-tailed Student’s *t* test, Mann–Whitney Test) were calculated using GraphPad Prism 7. For correlation, the Spearman rank correlation was used. *p* < 0.05 is considered statistically significant.

## Figures and Tables

**Figure 1 molecules-27-05191-f001:**
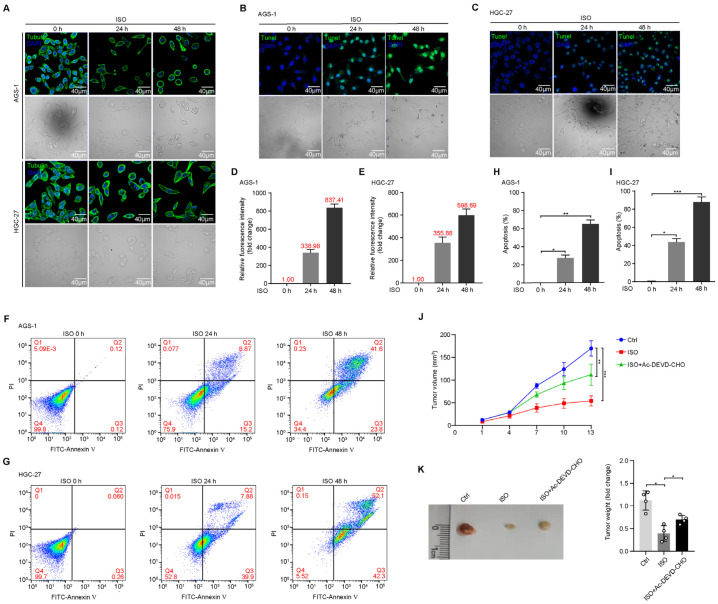
ISO induced apoptotic cell death of GC cells and suppressed tumor growth. ISO (20 μM) was used to treat cells at the indicated time. (**A**) Representative fluorescent images of AGS-1 and HGC-27 cells. Cytoskeleton (green) and nuclei (blue) were stained with α-Tubulin and DAPI, respectively. (**B**,**C**) TUNEL assays in AGS-1 (**B**) and HGC-27 (**C**) cells under ISO treatment. (**D**,**E**) Statistical analysis by calculating TUNEL-fluorescence intensity with ImageJ as shown in (**B**,**C**). (**F**,**G**) Flow cytometric analysis revealed the apoptosis of AGS-1 (**F**) and HGC-27 cells (**G**) with ISO treatment. (**H**,**I**) Statistical analysis of the apoptotic AGS-1 and HGC-27 cells shown in (**F**,**G**). (**J**,**K**) Volume (**J**) and weight (**K**) of xenografts with ISO and caspase-3 inhibitor treatment as indicated. * *p* < 0.05; ** *p* < 0.01; *** *p* < 0.001 calculated using unpaired Student’s *t* test (*n* = 3).

**Figure 2 molecules-27-05191-f002:**
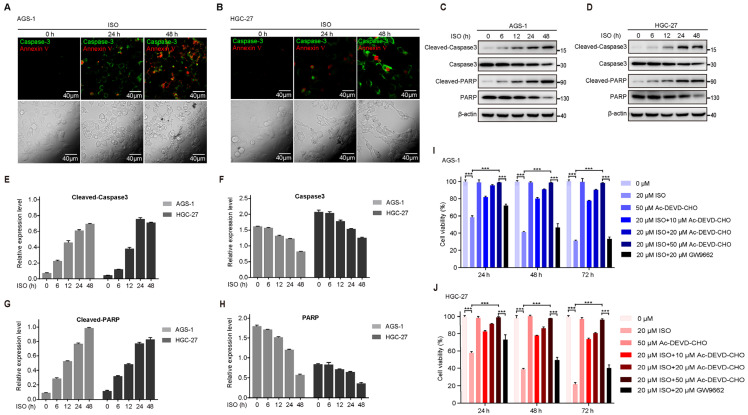
ISO treatment induces caspase-3 cascade in GC cells. Cells were treated with 20 μM ISO. (**A**,**B**) Representative images of AGS-1 (**A**) and HGC-27 (**B**) cells under annexin V/caspase-3 staining. (**C**,**D**) Cells were treated with ISO and harvested, and Western blotting was used to measure the expression of cleaved caspase-3, caspase-3, cleaved PARP, PARP, and *β*-actin in AGS-1 (**C**) and HGC-27 (**D**) cells. *Β*-actin was used as a loading control. (**E**–**H**) Statistics of the relative levels of proteins are shown in (**C**,**D**), as measured by ImageJ. (**I**,**J**) CCK-8 assays of AGS-1 (**I**) and HGC-27 (**J**) cells with or without 30-min Ac-DEVD-CHO, 2-h GW9662 pre-treatment. *** *p* < 0.001 calculated using unpaired Student’s *t* test (*n* = 3).

**Figure 3 molecules-27-05191-f003:**
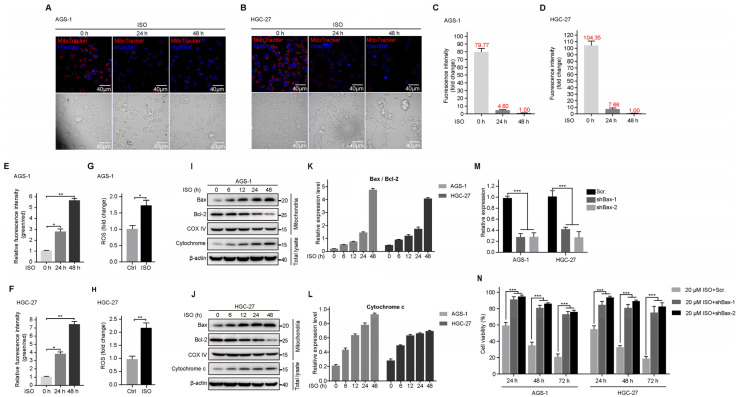
Mitochondria-dependent pathway mediates ISO-induced apoptosis of GC cells. Cells were treated with 20 μM ISO. (**A**,**B**) Representative images of two types of GC cells stained by Mito-Tracker Red CMXRos with the indicated ISO treatments. (**C**,**D**) Statistics of fluorescence intensity of Mito-Tracker Red CMXRos signals shown in (**A**,**B**). (**E**–**H**) MMP detected by JC-1 staining assays and ROS in two types of GC cells with the indicated ISO treatments. * *p* < 0.05; ** *p* < 0.01; compared to control (*n* = 3) by unpaired Student’s *t* test. (**I**,**J**) Expression of mitochondria-associated Bax, mitochondria-associated Bcl-2, COX IV, cytoplasmic cytochrome-c, and *β*-actin in AGS-1 (**I**) and HGC-27 (**J**) cells revealed by Western blotting. Bax, Bcl-2, and COX IV were detected with the mitochondrial fraction, while the others were detected with the total cell lysates. COX IV and *β*-actin were used as a loading control for mitochondria and total cell lysates, respectively. (**K**,**L**) Statistics of the relative ratio of Bax/Bcl-2 and expression level of cytochrome-c shown in (**I**,**J**). (**M**) Knockdown of Bax by shRNA in AGS-1 and HGC-27 cells, shown by RT–qPCR. *** *p* < 0.001 compared to Scr. (*n* = 3) are calculated using unpaired Student’s *t* test. (**N**) CCK-8 assays of Bax knockdown cells with 20 μM ISO treatment. *** *p* < 0.001 compared to Scr. calculated using unpaired Student’s *t* test (*n* = 3).

**Figure 4 molecules-27-05191-f004:**
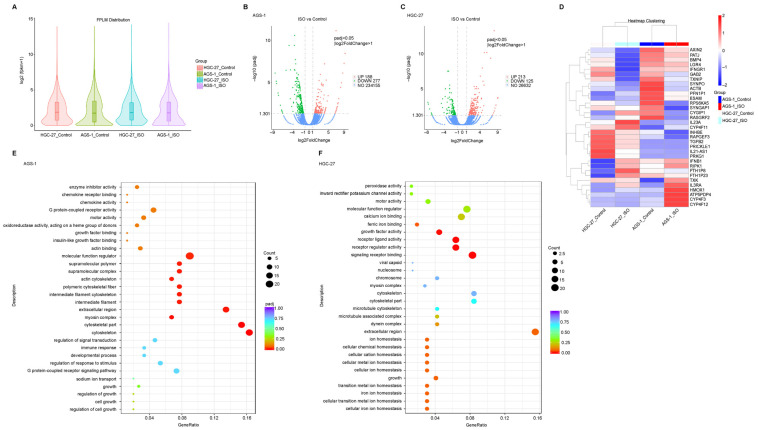
Transcript analysis of ISO-induced GC cells. (**A**) Violin plot shows the distribution of FPKM for each sample. (**B**,**C**) Volcano plot depicts the differentially expressed genes (DEGs) under ISO treatment of two types of GC cells. (**D**) DEGs of two types of GC cells revealed by heatmap. (**E**,**F**) GO annotation of the DEGs.

**Figure 5 molecules-27-05191-f005:**
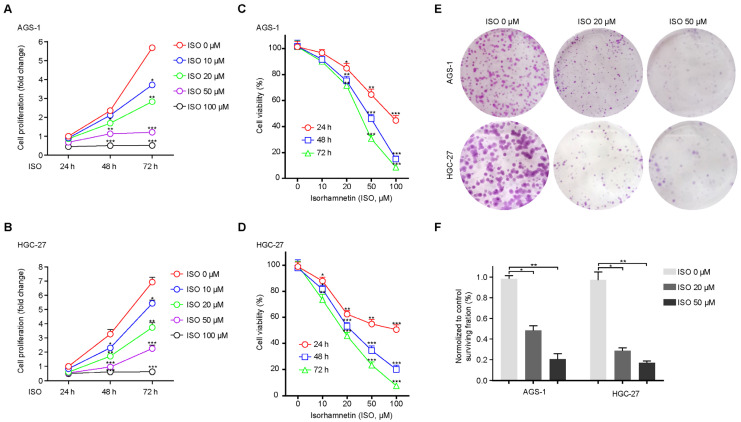
ISO inhibits GC cell proliferation. (**A**,**B**) AGS-1 and HGC-27 cells were treated with ISO (0–100 μM), and the cell proliferation was measured with CCK-8 assays at the indicated time points (24, 48, and 72 h). (**C**,**D**) Statistics of cell viability measured by CCK-8 assays in the two types of GC cells. (**E**,**F**) Colony formation and corresponding statistics of two types of GC cells under 24-h ISO pre-treatment (0, 20, and 50 μM). * *p* < 0.05; ** *p* < 0.01; *** *p* < 0.001 compared to ISO 0 μM at each time (**A**,**B**) or ISO 0 μM of each group (**C**,**D**) (*n* = 3) by Student’s *t* test.

**Figure 6 molecules-27-05191-f006:**
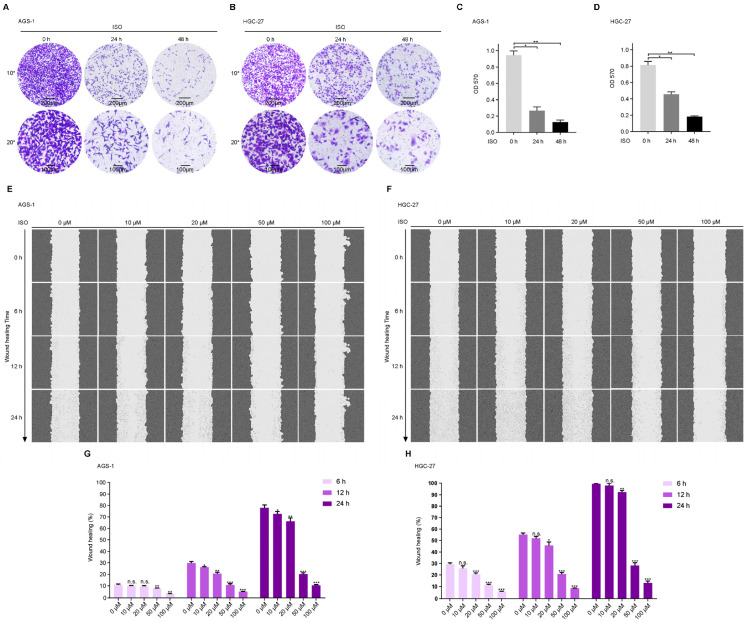
ISO suppressed migration of GC cells. (**A**,**B**) Transwell assays revealed the cell migration of the two types of GC cells under ISO (20 μM) treatment. (**C**,**D**) Statistics of cell migration efficiency of GC cells as shown in (**A**,**B**). (**E**,**F**) Wound-healing assay showed the migration of GC cells through 24-h continuous real-time imaging. (**G**,**H**) Statistical analysis of cells migration efficiency shown in (**E**,**F**) according to the gap area during indicated ISO treatment compared with the control. * *p* < 0.05; ** *p* < 0.01; *** *p* < 0.001; n.s. no significance compared to 0 h group (*n* = 3) by unpaired Student’s *t* test.

## Data Availability

The data used to support the findings of this study are available from the corresponding authors upon request.

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
