# Peer review of "Isorhamnetin Suppresses Human Gastric Cancer Cell Proliferation through Mitochondria-Dependent Apoptosis"

_molecules, 2022, doi:10.3390/molecules27165191_

Round 1

Reviewer 1 Report

Conclusions should be re-written in according to the abstract. New information should be included.

Author Response

Thanks for reviewer’s advice.   In the previous revised version, we have included description for all new data in figure 2 (line 137-143) and figure 3 (line 180-182).   As suggested, we supplement description for the new data in the discussion section to highlight the new information.   The revised part was marked in red.

Reviewer 2 Report

The authors have addressed all the concerns I raised on the previous draft. I recommend publication of the current draft.

Author Response

Many thanks for reviewer’s positive comments and constructive suggestion during the revision.

This manuscript is a resubmission of an earlier submission. The following is a list of the peer review reports and author responses from that submission.

Round 1

Reviewer 1 Report

Natural compounds are a very important source for the development of new anticancer drugs and treatments. Unfortunately, the authors were not the first ones who studied the suppression of the proliferation and migration of GC cells by Isorharmnetin. There is an article published on this topic in the Journal of Biological Chemistry in 2012 “Isorhamnetin Inhibits Proliferation and Invasion and Induces Apoptosis through the Modulation of Peroxisome Proliferator-activated Receptor Activation Pathway in Gastric Cancer” (JOURNAL OF BIOLOGICAL CHEMISTRY VOL. 287, NO. 45, pp. 38028 –38040, November 2, 2012). Although the published experiments and results in that paper are overlapping quite a bit with presented results, the authors didn’t reference it. The article can’t be published in the present format.  I would recommend the authors to revise their manuscript and highlight the novel results they obtained in comparison with that JBC article.
